# Planning for Sample Efficient Imitation Learning

**Zhao-Heng Yin**[*]   **Weirui Ye**[†]   **Qifeng Chen**[*]   **Yang Gao**[†‡§]

[*]HKUST    [†]Tsinghua University    [‡]Shanghai Qi Zhi Institute

## Abstract

Imitation learning is a class of promising policy learning algorithms that is free from many practical issues with reinforcement learning, such as the reward design issue and the exploration hardness. However, the current imitation algorithm struggles to achieve both high performance and high in-environment sample efficiency simultaneously. Behavioral Cloning (BC) does not need in-environment interactions, but it suffers from the covariate shift problem which harms its performance. Adversarial Imitation Learning (AIL) turns imitation learning into a distribution matching problem. It can achieve better performance on some tasks but it requires a large number of in-environment interactions. Inspired by the recent success of EfficientZero in RL, we propose EfficientImitate (EI), a planning-based imitation learning method that can achieve high in-environment sample efficiency and performance simultaneously. Our algorithmic contribution in this paper is two-fold. First, we extend AIL into the MCTS-based RL. Second, we show the seemingly incompatible two classes of imitation algorithms (BC and AIL) can be naturally unified under our framework, enjoying the benefits of both. We benchmark our method not only on the state-based DeepMind Control Suite, but also on the image version which many previous works find highly challenging. Experimental results show that EI achieves state-of-the-art results in performance and sample efficiency. EI shows over 4x gain in performance in the limited sample setting on state-based and image-based tasks and can solve challenging problems like Humanoid, where previous methods fail with a small amount of interactions. Our code is available at https://github.com/zhaohengyin/EfficientImitate.

## 1   Introduction

The real-world sequential decision process in robotics is highly challenging. Robots have to handle high dimensional input such as images, need to solve long horizon problems, some critical timesteps need highly accurate maneuver, and the learning process on the real robot has to be sample efficient. Imitation learning is a promising approach to solving those problems, given a small dataset of expert demonstrations. However, current imitation algorithms struggle to achieve these goals simultaneously. There are two kinds of popular imitation learning algorithms, Behavior Cloning (BC) and Adversarial Imitation Learning (AIL). BC formulates imitation learning as a supervised learning problem. It needs no in-environment samples, but it suffers from the covariate shift issue [37], often leading to test time performance degradation. Adversarial Imitation Learning (AIL) [16, 6] casts imitation learning as a distribution matching problem. Though AIL suffers less from the covariate shift problem and can perform better than BC on some domains, it requires an impractical number of online interactions [19, 22] and can perform badly on image inputs [34]. These drawbacks heavily limit its application in fields like robotics, where physical robot time matters. In summary, current imitation

---

[*]zhaoheng.yin@connect.ust.hk, cqf@ust.hk

[†]ywr20@mails.tsinghua.edu.cn, gaoyangiiis@tsinghua.edu.cn

[§]Corresponding author.

36th Conference on Neural Information Processing Systems (NeurIPS 2022).

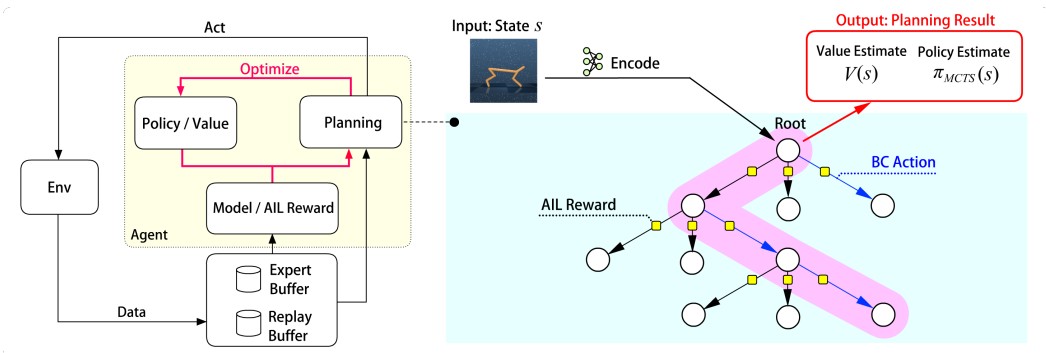

Figure 1: **Left:** The system-level overview of EfficientImitate. The agent (yellow area) takes actions in the environment and stores the data in the replay buffer. The data in the replay buffer and expert buffer are then used to train a model and AIL reward. The planning module then searches for improved policy and value in each state in the replay buffer, based on which the policy and value networks are optimized. **Right:** The planning procedure. We use a continuous version EfficientZero as our planner. We find that MCTS uniquely benefits from BC and can unify BC and AIL. For the expansion of each node, we sample actions from both the current policy (black arrow) and a BC policy (blue arrow). We use AIL reward (yellow cube) to encourage long-term distribution matching. MCTS searches (pink area) for best actions to maximize the cumulative AIL reward and update the estimated value and policy of the root node. The output of the planning procedure is the value estimate and the policy estimate of $s$, and the value and policy networks are optimized to fit them.

learning algorithms fail to achieve high online testing performance and high in-environment sample efficiency at the same time. Further, the two types of imitation algorithms seem to be incompatible since they are based on two completely different training objectives. Previous work finds that it is hard to unify them naively[32].

Inspired by the recent success in sample efficient RL, such as EfficientZero [48], we propose a planning-based imitation algorithm named EfficientImitate (EI) that achieves high test performance and high in-environment sample efficiency at the same time. Our method extends AIL to a model-based setting with multi-step losses under the MCTS-based RL framework. Our algorithm also unifies the two types of the previous imitation algorithms (BC and AIL) naturally, thanks to the planning component of our algorithm. Intuitively, BC gives a coarse solution that is correct most of the time but fails to match the expert's behavior in the long term. On the other hand, AIL knows the goal of the learning, i.e., matching state action distribution, but doesn't give the solution directly. Our method's planning component unifies those two methods and shows a significant performance boost, especially in the harder tasks such as Humanoid. We illustrate the detailed procedure in Figure 1.

We validate our idea not only on state-based tasks but also on image-based tasks, which relatively few previous algorithms can handle them [34]. EI achieves state-of-the-art results in sample efficiency and performance. It shows over 4x gain in performance in the limited sample setting on state-based and image-based tasks. On harder tasks such as Humanoid, the gain is even larger. A by-product of this paper is that we extend the recent EfficientZero algorithm to the continuous action space. We open-source the code at `https://github.com/zhaohengyin/EfficientImitate` to facilitate future research.

Our contributions in this paper are summarized as follows.

- We present EfficientImitate, a sample efficient imitation learning algorithm based on MCTS.

- We identify that MCTS can benefit from BC by using BC actions during the search, which is crucial for challenging tasks. EfficientImitate suggests a natural way to unify BC and AIL.

- We conduct experiments in both the state and the image domain to evaluate EI. Experimental results show that EI can achieve state-of-the-art sample efficiency and performance.

## 2 Related Work

### 2.1 Imitation Learning

Imitation learning (IL) aims to solve sequential decision-making problems with expert demonstration data. It has wide applications in games and robotics [4]. Compared with RL, one major benefit of IL is that it can avoid the notorious reward design problem. IL can be divided into two branches: BC [1] and IRL [31]. To solve the covariate shift problem of BC, researchers propose methods like dataset aggregation [37] and noise injection [24]. But these methods either require extra expert queries or exert constraints over the learning process. A recent variant branch of IRL is the Adversarial Imitation Learning (AIL) [16, 6]. AIL models IL as a state-action distribution matching problem. Many works extend AIL by using better distribution metrics such as $f$-divergence [50] and Wasserstein distance [5]. Though these methods can learn better reward functions and somewhat speed up the training, they do not directly focus on AIL's sample efficiency problem. Some works have drawn attention to sample efficiency, for example, [22, 38, 51] propose to use off-policy training in AIL training to reduce sample complexity. ValueDICE [23] reformulates AIL objective in an offline min-max optimization process, but recent work points out that it is an improved form of BC [28]. VMAIL [34] uses a model-based approach to improve sample efficiency. It collects latent rollout with a variational model and reduces online interactions. MoBILE [21] also shows a provably sample efficient model-based imitation approach. Compared with these methods, our method introduces MCTS planning to the off-policy imitation learning and uses it to unify BC and AIL to take advantage of both.

The idea of combining BC and AIL helps to improve the sample efficiency can be traced back to GAIL[16]. GAIL suggests using BC to initialize the policy network, but [32] finds that this does not work well because the initialized BC knowledge will be corrupted in AIL training. Then, [19] proposes to use an annealed (decaying) BC loss to regularize policy training to solve this problem. But in this work, we find that such a BC regularizer can be harmful to exploration when BC is incorrect. One concurrent work mitigates this issue by using adaptive BC loss and replacing AIL reward with optimal-transport-based imitation reward [13]. We also notice that AlphaGo [43] involves BC in their method, but they focus on RL rather than IL, and BC is only used to initialize the policy. Different from these methods, EI uses BC actions as candidates in MCTS.

### 2.2 Sample Efficiency in RL

The sample efficiency problem of imitation learning is closely related to that in RL. One line of work finds that the reward signal is not a good data source for representation learning in RL and is one reason for sample inefficiency. Then they utilize self-supervised representation learning to accelerate representation learning and improve the sample efficiency. Researchers propose to use contrastive learning [26], consistency-based learning [41, 49, 48], or pretrained representation [42] for this purpose. Some works also explore the possibility of applying self-supervised representation learning to imitation learning [4].

Another line of work focuses on RL with a learned model, which is promising for sample efficient learning [8, 10–12, 18, 27, 29, 48, 14]. These approaches usually imagine additional rollouts with the learned model or use it as a more compact environment representation for RL. Besides, some works also find that data augmentation can effectively improve sample efficiency [25, 46, 45]. EI also benefits from these approaches. It includes a model and applies representation learning to boost sample efficiency. In application, people also consider to augment RL with the demonstration [15, 35, 30, 20] to improve the sample efficiency. This can be viewed as a combination of RL and imitation learning and favor RL on real robots. We believe that our method can also be extended to this setting.

## 3 Background

### 3.1 Setting

We formalize the sequential decising making problem as Markov Decision Process $\mathcal{M} = (\mathcal{S}, \mathcal{A}, \mathcal{R}, \mathcal{T})$. Here, $\mathcal{S}$ is the state space, $\mathcal{A}$ is the action space, $\mathcal{R}$ is the reward function, and $\mathcal{T}$ is the transition dynamics. The agent's state at timestep $t$ is $s_t \in \mathcal{S}$. The agent takes action $a_t$ and

receives reward $r_t = \mathcal{R}(s_t, a_t)$. Its state at timestep $t + 1$ is then $s_{t+1} \sim \mathcal{T}(s_t, a_t)$. The objective of the agent is to maximize the return $\sum_{t=0}^{T} \gamma^t r_t$, where $\gamma$ is a discount factor.

In the imitation learning problem studied here, the agent has no access to the reward function $\mathcal{R}$ and transition dynamics $\mathcal{T}$. It is provided with a *fixed* expert demonstration dataset $\mathcal{D} = \{\tau_i\}$. Here, each $\tau_i = (s_0^E, a_0^E, s_1^E, a_1^E, ...s_T^E, a_T^E)$ is an expert trajectory that can achieve high performance in $\mathcal{M}$. The agent can not solicit extra expert demonstrations but can interact with the MDP, observing new states and actions, but not rewards. In this work, we define (in-environment) sample efficiency as the number of online interactions during training. We expect the agent to achieve high performance within a fixed online sample budget.

### 3.2 BC and AIL

BC considers imitation learning as a supervised learning problem. It trains a policy network $\pi$ to minimize the following loss function:

$$\mathcal{L} = -\mathbb{E}_{(s_i^E, a_i^E) \sim \mathcal{D}} \log \pi(a_i^E | s_i^E). \tag{1}$$

AIL treats imitation learning as a distribution matching problem. One typical AIL algorithm is GAIL. It trains a discriminator $D$ to distinguish the agent generated state-action tuple $(s_i, a_i)$ from those $(s_i^E, a_i^E)$ in the demonstration dataset by minimizing

$$\mathcal{L} = -\mathbb{E}_{(s_i^E, a_i^E) \sim \mathcal{D}, (s_i, a_i) \sim \rho} \left[ \log(D(s_i^E, a_i^E)) + \log(1 - D(s_i, a_i)) \right], \tag{2}$$

where $\rho$ is the state-action distribution induced by the agent. Meanwhile, it trains the agent to maximize the return with respect to the adversarial reward $r_t = -\log(1 - D(s_t, a_t))$ using any on-policy RL algorithm.

### 3.3 MuZero and its Extensions

Our planning method is based on MuZero [39] and its extensions. MuZero learns an environment model for MCTS. The model consists of an encoder network $f$, a dynamics network $g$, and a reward network $R$. It operates on abstract states [48]. Concretely, it gets the abstract state $h_t$ of the current state $s_t$ by $h_t = f(s_t)$. It can then predicts the future abstract states recursively by $h_{t+1} = g(h_t, a_t)$, and the rewards by $R(h_t, a_t)$. Besides the model, MuZero also contains a policy network and a value network. The policy network provides a prior over the actions at each node, and the value network calculates the expected return of the node. MuZero uses the model, the policy network, and the value network to search for improved policy and value for each state with MCTS. We refer the readers to the original MuZero paper for details.

**Sampled MuZero**  Sampled MuZero [17] extends MuZero from the discrete action domain to the continuous action domain, which is of our interest in this paper. At each node $s$ to expand, Sampled MuZero samples $K$ actions $\{a_i\}_{i=1}^{K}$ from current policy $\pi(a|s)$. During the search, it selects action $a^*$ from the sampled actions that maximize the probabilistic upper confidence bound

$$a^* = \arg \max_{a \in \{a_i\}} Q(s, a) + c(s)\hat{\pi}(a|s) \frac{\sqrt{\sum_b N(s, b)}}{1 + N(s, a)}, \tag{3}$$

where $\hat{\pi}(a|s) = \frac{1}{K} \sum_i \delta(a, a_i)$. $Q(s, a)$ is the current $Q$-estimation of the pair $(s, a)$. $N(s, a)$ denotes the times that this pair is visited in MCTS. $c(s)$ is a weighting coeffcient. During policy optimization, MuZero minimizes the Kullback-Leibler divergence between the current policy $\pi$ and the MCTS statistics $\pi_{\mathrm{MCTS}}$ at the root node $D_{KL}(\pi_{\mathrm{MCTS}} || \pi)$.

**EfficientZero**  We also apply EfficientZero [48] in this paper. EfficientZero improves the sample efficiency of MuZero by using a self-supervised representation learning method to regularize the hidden representation. It uses a SimSiam-style structure [3] to enforce the similarity between the predicted future representation and the real future representation.

## 4 EfficientImitate

In this section, we present our EfficientImitate algorithm. We first present an MCTS-based approach to solving the AIL problem in Section 4.1. Then we show a simple yet effective method to unify BC

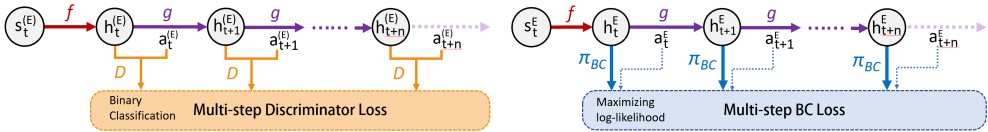

Figure 2: Computation flow of loss functions. **Left:** Multi-step Discriminator Loss. We do not distinguish between the calculation for expert and agent here, and use a superscript $(E)$ to indicate that the computation applies to both. **Right:** Multi-step BC Loss. It applies to the expert sequences.

and AIL with MCTS in Section 4.2. We briefly discuss the implementation in Section 4.3, and the full details can be found in the Appendix.

## 4.1 Extending AIL to MCTS-based RL

Traditionally, the adversarial imitation learning (AIL) algorithm trains a discriminator $D$ between the policy samples and the expert samples and uses some form of $D$, such as $-\log(1 - D)$, as the reward function. Then some model-free RL algorithms are used to maximize the cumulative reward. In MCTS-based RL algorithms, such as MuZero [39] and EfficientZero [48], the reward function is used in the value target computation and the MCTS search. The use in value target computation is similar to prior model-free RL algorithms, where the value target is computed with n-step value bootstrapping on the actual observations. However, during the MCTS search, the rewards are computed on the abstract state obtained by running the forward dynamics function $h_{t+1} = g(h_t, a_t)$ multiple times. If we were training the discriminator only on actual observations of the expert and the policy rollouts, the discriminator might not generalize well to abstract states outputted by the forward dynamics functions. Therefore, we train the discriminator with the model-based rollout. Specifically, we sample sequence $(s_t, a_{t+1}, ..., a_{t+n})$ in replay buffer $\mathcal{B}$ and expert sequence $(s_{t'}^E, a_{t'}^E, a_{t'+1}^E, ..., a_{t'+n}^E)$ in demonstration dataset $\mathcal{D}$ and minimizes following multi-step discriminator loss function:

$$\mathcal{L}_D = -\mathbb{E}_{(s_t^E, a_{t:t+n}^E) \sim \mathcal{D}, (s_t', a_{t:t+n}') \sim \mathcal{B}} \left[ \sum_{i=0}^{n} \log(D(h_{t+i}^E, a_{t+i}^E)) + \log(1 - D(h_{t'+i}, a_{t'+i})) \right]. \quad (4)$$

Here, $h_{t+i}$ (and $h_{t'+i}$) terms are produced by the forward dynamics in EfficientZero (Figure 2). We use the GAIL transition reward $R(h, a) = -\log(1 - D(h, a))$, and then the MCTS planner searches for action that can maximize cumulative GAIL reward to guarantee long-term distribution matching. Note that V-MAIL also propose a similar discriminator training technique, but under the Dreamer [11] model.

Besides, since the discriminator's input is based on the representation rather than raw input, the discriminator should be trained with the encoder jointly. However, this can lead to a highly non-stationary reward during the bootstrap value calculation. To mitigate this issue, we also propose to use a target discriminator for bootstrap value calculation. This can make the training more stable.

Though we use the GAIL reward here, one can also use other kinds of AIL and IRL reward functions proposed in recent research. Using the GAIL reward can already achieve satisfactory performance in our experiments. When the real reward presents, one may also combine this into planning [20]. This may favor application scenarios where handcrafting a reward function is not hard. We do not study this case here and leave it to future work.

## 4.2 Unifying BC and AIL in MCTS

As discussed in related work, researchers realize that using BC can improve AIL's sample efficiency by providing a good initial BC policy or constraining the policy to BC. However, these existing solutions are not good enough in practice. The main pitfall in these methods is that BC knowledge in the policy network is destined to be forgotten if the policy network is trained with the AIL objective, and then BC will no longer be helpful [32].

We observe that MCTS can naturally unify the BC and AIL methods, enjoying the benefit of both and being free from this pitfall. We propose to plug BC actions into MCTS as candidates at each

node and use a planning process to search for an improved solution. This time, the BC actions are consistently considered throughout the entire training procedure without being forgotten. Concretely, we train a BC policy $\pi_{BC}$ and use a mixture policy $\tilde{\pi}$ for the sampling at each node in MCTS:

$$\tilde{\pi} = \alpha\pi_{BC} + (1-\alpha)\pi. \tag{5}$$

$\alpha$ is a mixture factor, which is fixed during training and $\pi$ is the current policy. We use $\alpha = 0.25$ in this paper. This ensures that a small fraction of action samples are from the BC policy. During planning, the BC actions are evaluated and will be frequently visited and selected as output if they can lead to long-term distribution matching. This can then reduce the effort of finding good expert-like actions from scratch as desired. Moreover, another unique advantage of this procedure is that it does not fully trust BC like [19], which forces the output of the policy network to be close to BC. When BC is wrong due to covariate shifts or insufficient demonstrations, it can neglect these BC actions and allow the policy to search for better actions. This ensures that BC does not hurt training. However, due to the conceptual simplicity, one arising question is whether this approach can be applied to other model-based methods. Here, we take Dreamer [11] as an example. Though Dreamer builds a model of the environment, it only uses the model to roll-out the policy for policy optimization. In other words, the model is not used to evaluate whether a specific BC action is good or not in the long term, so our idea can not be applied directly to Dreamer. From this example, we see that the core of our idea is to leverage the power of planning, only with which the long-term outcomes of certain (BC) actions can be calculated.

For the training of $\pi_{BC}$, we minimize the following multi-step BC objective (Figure 2):

$$\mathcal{L}_{BC} = \mathbb{E}_{(s_{t'}^E, a_{t':t'+n}^E)\sim\mathcal{D}} \left[ \sum_{i=0}^{n} -\log(\pi_{BC}(a_{t'+i}^E | h_{t'+i}^E)) \right]. \tag{6}$$

This is to avoid distributional shifts during multi-step prediction in MCTS. For the training of the policy, we still minimize $D_{KL}(\pi_{\mathrm{MCTS}}||\pi)$.

Note that the BC design proposed here is not coupled with AIL. It can go beyond imitation learning and be applied in other robot learning settings, such as RL with demonstration [35].

## 4.3 Implementation

We first implement a continuous version EfficientZero for planning, and the details can be found in the Appendix. The BC policy network is simply a duplicate of the policy network. The discriminator and BC policy networks share the same encoder network with the policy network and value network. The overall loss function for optimization is

$$\mathcal{L} = \mathcal{L}_{EZ} + \lambda_d\mathcal{L}_D + \lambda_{bc}\mathcal{L}_{BC}. \tag{7}$$

$\mathcal{L}_{EZ}$ is EfficientZero's loss function (excluding reward loss). All the networks are trained jointly to minimize this loss function 7. We use the Reanalyze algorithm [40, 48] for offline training, and we require that all the samples should be reanalyzed.

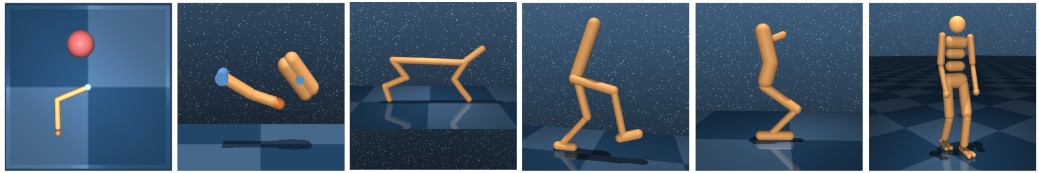

Figure 3: Part of the tasks used in our experiments. From left to right: Reacher, Finger Spin, Cheetah Run, Walker Walk, Hopper Hop, Humanoid Walk.

## 5 Experiments

In this section, we evaluate the sample efficiency of the proposed method. We measure the sample efficiency by evaluating the performance of an algorithm at a small number of online samples. We also analyze the effect of the BC actions and planning.

Table 1: Evaluation result on the state-based DeepMind Control Suite. We use the average score on three random seeds. Our method can achieve the state-of-the-art result compared with the baselines.

| Task | Cartpole | Ball | Reacher | Finger | Cheetah | Walker | Hopper | Humanoid |
|------|----------|------|---------|--------|---------|--------|--------|----------|
| **Budget** | 10k | 10k | 50k | 50k | 50k | 50k | 50k | 500k |
| BC | 0.59 | 0.44 | 0.83 | 0.76 | 0.58 | 0.16 | 0.03 | 0.11 |
| DAC | 0.13 ±0.12 | 0.18 ±0.01 | 0.22 ±0.02 | 0.53 ±0.05 | 0.33 ±0.04 | 0.26 ±0.04 | 0.00 ±0.00 | 0.01 ±0.00 |
| ValueDICE | 0.21 ±0.01 | 0.23 ±0.01 | 0.15 ±0.01 | 0.04 ±0.01 | 0.50 ±0.08 | 0.54 ±0.09 | 0.03 ±0.00 | 0.00 ±0.00 |
| SQIL | 0.23 ±0.01 | 0.27 ±0.05 | 0.21 ±0.02 | 0.02 ±0.00 | 0.05 ±0.01 | 0.11 ±0.03 | 0.24 ±0.10 | 0.06 ±0.01 |
| Ours | **0.98** ±**0.01** | **0.99** ±**0.01** | **0.90** ±**0.04** | **0.99** ±**0.00** | **0.96** ±**0.02** | **1.03** ±**0.01** | **0.92** ±**0.02** | **0.74** ±**0.04** |

Table 2: Evaluation result on the image-based DeepMind Control Suite. We use the average score on three random seeds. Our method can achieve state-of-the-art results compared with the baselines.

| Task | Cartpole | Ball | Finger | Cheetah | Reacher | Walker | Hopper |
|------|----------|------|--------|---------|---------|--------|--------|
| **Budget** | 50k | 50k | 50k | 50k | 100k | 100k | 200k |
| BC | 0.30 | 0.32 | 0.14 | 0.37 | 0.26 | 0.15 | 0.02 |
| DAC | 0.08 ±0.01 | 0.26 ±0.02 | 0.00 ±0.00 | 0.04 ±0.01 | 0.25 ±0.05 | 0.10 ±0.02 | 0.01 ±0.00 |
| ValueDICE | 0.18 ±0.02 | 0.27 ±0.02 | 0.01 ±0.00 | 0.06 ±0.01 | 0.15 ±0.02 | 0.08 ±0.00 | 0.00 ±0.00 |
| SQIL | 0.26 ±0.03 | 0.77 ±0.05 | 0.00 ±0.01 | 0.06 ±0.00 | 0.36 ±0.04 | 0.32 ±0.05 | 0.04 ±0.02 |
| VMAIL | 0.57 ±0.03 | 0.61 ±0.11 | 0.06 ±0.03 | 0.13 ±0.04 | 0.34 ±0.02 | 0.24 ±0.07 | 0.07 ±0.04 |
| Ours | **0.94** ±**0.02** | **0.93** ±**0.01** | **1.00** ±**0.01** | **0.92** ±**0.01** | **0.86** ±**0.06** | **0.98** ±**0.01** | **0.70** ±**0.01** |

## 5.1 Setup

We use the DeepMind Control Suite [44] for evaluation. We use the following tasks: Cartpole Swingup, Reacher Easy, Ball-in-cup Catch, Finger Spin, Cheetah Run, Walker Walk, Hopper Hop, and Humanoid Walk. We conduct both state-based and image-based experiments. Note that many previous imitation learning works use the OpenAI Gym [2] version of these tasks for evaluation. We find that the DMControl version used here brings extra challenges by using more challenging initial states. Take the Walker task as an example; the initial state in OpenAI Gym is standing. However, in DMControl, the agent's initial state is lying on the ground, and the agent should also learn to stand up first from very limited data. For the state-based experiments, we allow 10k-50k online steps in the environment based on the difficulty of each task. Since learning a robust and meaningful visual representation requires more data for image-based experiments, we allow 50k-100k online steps. Detailed setup will be shown in the result. We train SAC [9] policies to collect expert demonstrations for imitation learning. The expert demonstrations are not subsampled. We use 5 demonstrations in the state-based experiment, except for Reacher and Humanoid, where we use 20 demonstrations. We use 20 demonstrations in the image-based experiments.

## 5.2 Baselines

We present several baselines of sample efficient imitation learning. (1) **DAC** DAC [22] is an adversarial off-policy imitation learning method. It matches the distribution of the replay buffer and that of the expert demonstration dataset using the TD3 [7] algorithm. (2) **SQIL** SQIL [36] is a non-adversarial off-policy imitation learning method. It labels all the expert transitions with reward 1 and non-expert transitions with reward 0. Then it trains a SAC policy over these relabeled data. SQIL is a regularized form of BC. (3) **ValueDICE** ValueDICE [23] considers imitation learning as a distribution matching problem and solves it with a min-max optimization process. (4) **VMAIL** VMAIL [34] is a model-based visual imitation learning method. It learns a variational model for simulating on-policy rollouts. We only evaluate VMAIL on the image-based domain, as they did in

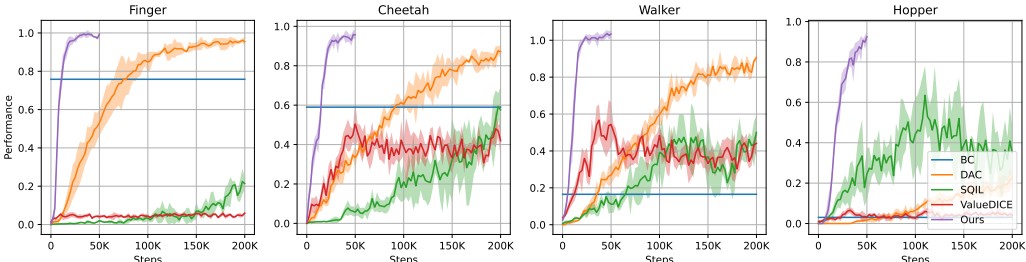

Figure 4: The performance curve on the state-based tasks. The results are averaged over three seeds. The shaded area displays the range of one standard deviation.

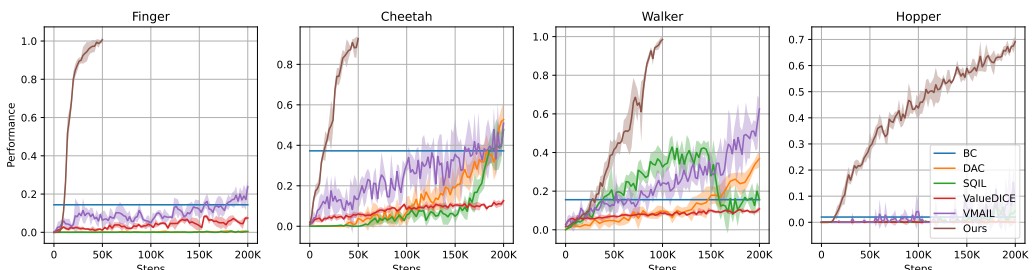

Figure 5: The performance curve on the image-based tasks. The results are averaged over three seeds. The shaded area displays the range of one standard deviation.

the original paper. Besides the online imitation learning baselines, we also include BC as an offline baseline.

## 5.3 Results

**State-based experiments**    Table 1 shows the state-based experiments' evaluation results within the given budget. We also plot the performance curve of four challenging tasks in Figure 4. The performance is normalized to 0.0 and 1.0 with respect to the performance of the random agent and the expert. We find that our proposed method can outperform all the baselines by a large margin. Except for BC, these baseline methods could hardly learn meaningful behaviors using a limited online budget. We find that DAC is a strong baseline. Its performance can grow to the expert in 200k samples in most tasks except Hopper, where it will eventually get stuck. Our method is much more sample efficient than the best of these baseline methods. For some tasks like Walker Walk and Cheetah Run, our method only requires about 20k steps to reach near-expert performance, equivalent to 80 online trajectories (around 0.5 hours in real). This result is notable for the robotics community. It shows that online imitation learning is possible with only a handful of trials, and applying it directly on a real locomotion robot is possible.

**Image-based experiments**    So far, image-based tasks are still challenging for adversarial imitation learning algorithms, and the evaluation of most of the prior AIL works is carried out in the state-based tasks. Table 2 shows the evaluation result within the given budget in the image-based experiments (see Figure 5 for curves). Our method can also learn expert behavior given a slightly larger budget. Still, most of the baselines fail to match experts' behavior using the given budget. We notice an inherent difficulty in learning a robust visual representation for adversarial training in the limited data set in image-based tasks. Discriminators can judge whether a behavior is expert-like using various possible features in this case. Solving this open problem is out of the scope of this paper. In the presence of such a difficulty, EI can still achieve good sample efficiency in most of the tasks.

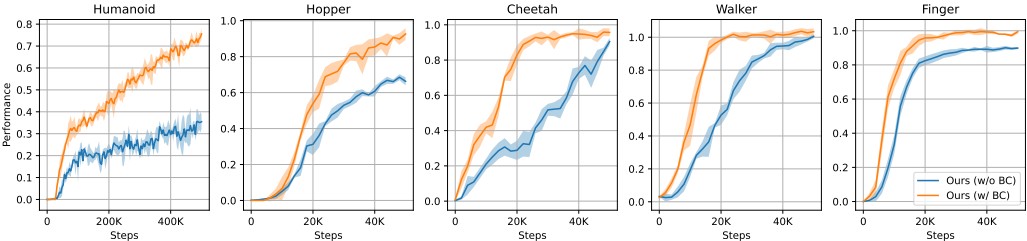

Figure 6: The performance curves of our method with and without BC actions at each expansion. The plots are sorted according to the difficulty of the corresponding task. The leftmost one is the most difficult task, Humanoid. The results are averaged over three seeds. BC actions have a great impact on the sample efficiency and performance.

## 5.4 Analysis

**Effect of BC**    We carry out ablation analysis on the BC to see whether it helps in our method. We set $\alpha = 0$ to remove all the BC actions and see how our method's performance and sample efficiency will change. The result is shown in Figure 6. We find that the performance of our method degrades after we remove all the BC actions in MCTS. The effect is task-specific, and we find BC is more helpful in those challenging tasks. For example, in tasks that are high-dimensional or have a hard exploration process like Humanoid and Hopper, removing BC actions will make the learning process stuck in local minima. At that local minima, the agent struggles to find the correct action that matches the expert's behavior. Although removing BC actions does not trap the agent in local minima in some relatively simpler tasks like Cheetah and Walker, it slows down the training. It doubles the number of online interactions to reach expert performance. This result confirms that using BC actions can indeed provide a good solution to the distribution matching problem, which can help to speed up learning and improve performance. We also notice that even when we remove the BC actions, the method is still able to outperform the previous baselines; this suggests that planning with AIL alone is also powerful.

**Other Ways to use BC**    We then study another two variants of using BC: (1). **BC-Ann**. This variant does not use BC actions in MCTS but exerts an annealing BC loss to the policy network like [19]. (2). **BC-Rep**. This variant does not use BC actions in MCTS but still uses BC loss to regularize the representation. We test these variants on the Humanoid Walk (Figure 7). We find that these variants do not lead to an essential improvement. For BC-Ann, it harms the performance in the early stage (before 100k) since the BC regularization will constrain the agent's policy near the BC policy, which contains an error and hinders learning. The agent only starts to acquire meaningful behav-

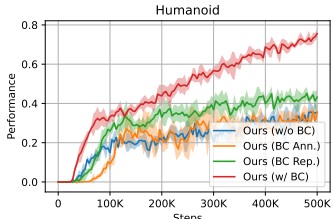

Figure 7: Results of different ways of using BC.

ior after the regularization decays, but at that time, BC does not help much and can not lead to improvement. Compared with BC-Ann, BC-Reg is more helpful here. This is possibly because BC-Reg makes the encoder focus on more useful features. However, BC-Reg still gets stuck in a local minimum. This result suggests that using BC actions directly for exploration can be essential for improving AIL. Using BC simply as a regularizer may not be the ideal approach though it can be useful sometimes.

**Ablation of Planning**    In this part, we study to what extent planning can help to learn and whether insufficient search in MCTS leads to inferior performance. We study $K$, the number of sampled actions at each node, and $N$, the number of simulations. The default value of $K$ and $N$ in the previous experiments are 16 and 50. We sweep $K \in \{4, 8, 16, 24\}$ and $N \in \{5, 10, 25, 50\}$ to evaluate their effects. We collect the result on the state-based Cheetah, Walker, and Hopper task and report the averaged relative performance change at the given budget used in previous experiments (see Table 3). The general trend is that larger $K$ and $N$ lead to better imitation results. We find that varying $K$ only affects the performance a little, and $K = 4$ can also work well. Compared with $K$, $N$ has a larger

impact. When the number of simulations becomes small, the performance drops significantly. This result also explains why we can achieve a large improvement over DAC even without BC.

Table 3: Ablation of planning. We report the relative change of performance at the given budget.

| **Param** | $K = 4$ | $K = 8$ | $K = 16$ | $K = 24$ | $N = 5$ | $N = 10$ | $N = 25$ | $N = 50$ |
|---|---|---|---|---|---|---|---|---|
| Result | $-8.4\%$ | $-3.2\%$ | $0.0\%$ | $1.1\%$ | $-27.3\%$ | $-15.5\%$ | $-4.6\%$ | $0.0\%$ |

## 6  Discussion

In this paper, we presented EfficientImitate, an MCTS-based imitation learning algorithm. We extended AIL to a model-based setting and solved it with MCTS in a sample-efficient way. Moreover, we proposed a method to unify BC and AIL in MCTS, enjoying the benefits of both. Experimental results in state-based and image-based tasks showed that EfficientImitate can achieve state-of-the-art sample efficiency and performance.

**Limitations**   One limitation of this work is that the computation process of MCTS is more expensive compared with that of the model-free methods, though this is a common issue of model-based methods. One possible approach to mitigate this issue can be using better MCTS acceleration methods [47]. Besides, in this paper, we did not study the long horizon problem with multiple objects, which is a common case in robotic manipulation. However, this requires the model to predict the interaction with multiple objects, which is still a challenging open problem in the learning community [33] and orthogonal to our contribution. We believe that our framework can be combined with the works in this field to handle this challenge.

**Future Work**   There are many problems to study along our direction. First, since we only use the vanilla AIL algorithm here, it is interesting to see if using more advanced algorithms such as optimal-transport-based learning [5] will make our algorithm more powerful. Second, due to the modularity of our method, one can try to extend EfficientImitate to more general settings like RL with demonstration, which will also favor the application scenarios. Third, in this work we consider an online learning setting, one possible future direction is to study the use of EfficientImitate on the existing offline interaction dataset to further reduce the dependence on in-environment samples.

In conclusion, we believe that this work shows a promising direction and opens up new possibilities for model-based methods in robot learning.

## Acknowledgments and Disclosure of Funding

This work is supported by the Ministry of Science and Technology of the People's Republic of China, the 2030 Innovation Megaprojects "Program on New Generation Artificial Intelligence" (Grant No. 2021AAA0150000). This work is also supported by a grant from the Guoqiang Institute, Tsinghua University.

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
