# OpenReview forum: "Planning for Sample Efficient Imitation Learning"
_NeurIPS.cc/2022/Conference — NeurIPS 2022 Accept_

### Official Review · Reviewer_q56x · 2022-07-10

**Rating:** 7
**Confidence:** 4
**Soundness:** 3 good
**Presentation:** 3 good
**Contribution:** 3 good

**Summary:**

This paper presents EfficientImitate (EI), which combines Adversarial Imitation Learning (AIL) with model-based planning with MCTS, for sample-efficient imitation learning. During training, an expert buffer is maintained in addition to the standard replay buffer used in EfficientZero. A discriminator is used to predict rewards by contrasting expert demonstrations and agent replays. Next, MCTS is performed each time step using the predicted reward from AIL. In addition, EI unifies BC with AIL by directly including BC actions in the MCTS, which significantly improves the performance. Experiments suggest that EI has achieved significantly better performance than the SOTA methods in various continuous control tasks.


**Questions:**

1/ What are the failure modes of the proposed method?

2/ Can the learned reward signal and state embeddings be interpreted in some way?

3/ How are the reward, value function, and action distribution represented in the MCTS? There are different implementations for Sample Muzero. Some of them are using distributional rewards and values, with discretized action spaces, while others are using continuous actions, e.g., Gaussians, and searching with sampled actions. In addition, how do these design choices affect the final performance of EI?

**Limitations:**

I think one limitation of work is that for lots of continuous control tasks, inference speed is critical. However, MCTS is a slow process. Although it achieves a very good performance during evaluation, it might not be appropriate for real-world situations, where inference speed is critical.

Overall, I still think this is a valuable paper and would like to vote for acceptance.

**Strengths And Weaknesses:**

Strengths:
Overall, the paper is well-written and easy to follow. Solid experiments have been conducted to support the claims made by the paper. The proposed idea has achieved a very strong performance in experiments, outperforming the state-of-the-art (SOTA) methods by a large margin, with much fewer environment interactions.

There are some minor weaknesses of the paper.

1/ The work mainly focuses on the continuous control domains, which has a relatively dense reward and limited planning horizon. However, MCTS is naturally designed for tasks with discrete action spaces, e.g., the game of Go or Atari games, and they are not considered during the experiments. My feeling is that with a sparse reward and long-planning horizon, learning reward from AIL might not be sufficient and may cause extra issues to MCTS. In that case, balancing BC and AIL will be a tricky issue.

2/ The proposed idea is relatively straightforward: it is a simple combination of AIL and MCTS. I think it would be better to dig deeper into the proposed framework.

---

> ### Author Response · Authors · 2022-08-02
> **Thanks for the review!**
>
> Thank you for reviewing our paper! We are glad that you find our paper valuable. We address your questions below.
>
> > Q1. What are the failure modes of the proposed method?
>
> So far, we have not identified failure modes for the state-based tasks. (Though our method gets 0.74 expert performance at 500k env samples for the Humanoid task, it can actually get 0.92 expert performance at 800k, at which all the baselines still learn nothing (< 0.20 expert performance)). In the image-based tasks, however, we find that the proposed method can still fail when the discriminator overfits to wrong, spurious features in the image (Line 265-268). For example, it would over-penalize the agent when it does not match some irrelevant details. However, this is a common problem for adversarial imitation learning, and solving it is orthogonal to our main contribution.
>
> > Q2. Can the learned reward signal and state embeddings be interpreted in some way?
>
> Yes. The reward signal can suggest whether the agent’s behavior looks like the expert's behavior during training. When the agent chooses actions that more resemble the expert’s actions, it will receive a higher reward. For the state embeddings, one approach to interpret it is by the t-SNE plot. We use the image-based Walker experiment as an example (see Appendix E). We use the trained model at 100k env steps to generate the state embeddings of one expert trajectory, and in environment trajectory at 0k, 25k, 50k, 75k, and 100k steps. Then we use t-SNE to visualize the embeddings on the 2D plane. As is shown in the figure, the agent’s trajectory gradually matches expert’s trajectory (blue) during training. Moreover, the expert’s trajectory has a circle structure, which represents the periodic pattern of the Walker’s walking behavior. Therefore, our model can represent the environment in a meaningful way.
>
>
> > Q3. How are the reward (logits), value function (dis.), and action distribution represented in the MCTS? There are different implementations for Sample Muzero. Some of them are using distributional rewards and values, with discretized action spaces, while others are using continuous actions, e.g., Gaussians, and searching with sampled actions. In addition, how do these design choices affect the final performance of EI?
>
> The reward is represented by logits with sigmoid (same as the GAIL). The value function is represented by a discretized categorical distribution as in MuZero. The action distribution is represented by a Gaussian distribution followed by a tanh function. The policy neural network outputs the mean and diagonal std of the Gaussian distribution. We follow the design choices in EfficientZero and Sampled Muzero. We hypothesize that the different design choices in our case may have a similar effect to that in Sampled MuZero.

---

> > ### Comment · Reviewer_q56x · 2022-08-07
> > **Vote for poster acceptance**
> >
> > Thanks for the response. The additional discussions and visualizations have addressed my questions. Nevertheless, as discussed in my original review, I am still concerned if a good reward / discriminator can be learned in tasks with long-planning horizon and sparse reward.
> >
> > Overall, I think this is a good paper and I would vote for poster acceptance.

---

### Official Review · Reviewer_t1UE · 2022-07-11

**Rating:** 5
**Confidence:** 3
**Soundness:** 3 good
**Presentation:** 3 good
**Contribution:** 3 good

**Summary:**

This paper proposes a new IL called EfficientImitate (EI), a planning-based imitation learning method that can achieve high sample efficiency and performance. EI extends AIL to a model-based setting and solve it with MCTS with high sample-efficiency. EI can benefit from BC by using the BC actions in the search. EI provides a natural way to unify the two classes of existing IL methods, BC and AIL. EI uses BC actions as candidates in MCTS. The performance of EI in both state-based and image-based control tasks are amazingly good.

**Questions:**

1. The base method, MuZero and EfficientZero, also works well in discrete action space. How is the performance of EI in discrete action space?

2. EI uses the Reanalyzed algorithm for offline training, and requires all the samples should be reanalyzed. What is the motivation for using the Reanalyzed algorithm here, and how significant does it contribute to the sample-efficiency of EI? Have you used the Reanalyzed algorithm for other baselines?

3. How to choose the balancing weights $\lambda_d$ and $\lambda_{bc}$ in Eq. (7)?

**Limitations:**

1. The performance of EI with discrete action space is unknown.

2. There is no discussion on limitations and potential social impacts.

**Strengths And Weaknesses:**

==Originality==

The proposed EI is novel and interesting. The main idea is to extend AIL to a model-based setting with multi-step losses under the MCTS-based RL framework. Thanks to the planning component of the algorithm, EI naturally unifies two types of previous imitation learning methods (BC and AIL), and shows a significant performance boost. The connections and differences of this work and previous work are well discussed and the related work are cited in the paper.

==Quality==

The proposed method is technically sound. The experiments are conducted in diverse settings with ablations, and the results well supports the claims. Although there are some limitations, this work is a complete piece of work in planning-based IL.

==Clarity==

This paper is clearly written and well organized. The problem is well formulated. The method is clearly motivated and introduced. The limitations and potential impacts are also discussed.

==Significance==

This paper provides an novel sample-efficient IL method that extends AIL into the MCTS-RL.

---

> ### Author Response · Authors · 2022-08-02
> **Thanks for the review!**
>
> Thank you for reviewing our paper! We address your questions and concerns below.
>
> > Q1. The base method, MuZero and EfficientZero, also works well in discrete action space. How is the performance of EI in discrete action space?
>
> We have just carried out experiments in the LunarLander-v2 environment provided by the OpenAI gym, where the agent uses four actions to land a spacecraft. We collect 5 expert trajectories for imitation learning. We follow the task setup in SQIL. Concretely, in the expert trajectories, the spacecraft (expert agent) is initialized at a fixed initial position. But during training and evaluation, the initial position of agent is perturbed so that the agent should learn to deal with different circumstances. We provide the agent with 5 expert trajectories, and allow 50k in-env samples for training. We find that our method could solve this task successfully. The results and comparisons are as follows.
> |  Method   | Performance |
> |  -------  | ----  |
> | BC | 0.76 |
> | DAC* | 0.32 ± 0.05 |
> | SQIL | 0.80 ± 0.03 |
> | ValueDICE | 0.71 ± 0.04 |
> | Ours | **0.90 ± 0.03** |
> *Note that DAC is based on TD3, which is not designed for the discrete action space. We make DAC’s extension to the discrete action space by ourselves.
>
> Therefore, our method is also effective in the discrete action space. Due to the limited resource and time during the rebuttal phase, we do not study more complex discrete tasks here. However, we believe that our method can also work well in those tasks.
>
> For the implementation of EI in this case, we first let the policy network (and the BC policy network) output a categorical distribution, which represents the probability of taking each discrete action. During recurrent inference, we turn the discrete action into an one-hot vector, concatenate it with the current state, and feed them to the dynamics network and discriminator to calculate the next state and the AIL reward. The other parts remain the same.
>
> > Q2. EI uses the Reanalyzed algorithm for offline training, and requires all the samples should be reanalyzed. What is the motivation for using the Reanalyzed algorithm here, and how significant does it contribute to the sample-efficiency of EI? Have you used the
> Reanalyzed algorithm for other baselines?
>
> Thanks for the question. Since the Reanalyze algorithm is an optional add-on component proposed by the original MuZero algorithm, we mention the use of Reanalyze algorithm to highlight that it has now been used as a standard component in the recent EfficientZero algorithm and here. In MuZero/EfficientZero, the motivation of the Reanalyze algorithm is to improve the performance and sample efficiency by providing a more accurate value target and policy target for the past visited states with MCTS. However, the sample efficiency of EI also comes from the BC component besides the Reanalyze algorithm. BC is very crucial here because it directly points out the potential right action in high-dimensional action spaces, without which even extensive MCTS can be not so efficient. After we remove the action provided by BC, the sample efficiency of EI can drop by half. EI gets stuck in local minima in challenging tasks without BC (Figure 6). Therefore, the Reanalyze only partly contributes to the sample efficiency of EI.
>
> Since the Reanalyzed algorithm is coupled with MuZero-style MCTS, it cannot be used for the other model-based imitation baselines directly (they are not based on MCTS). However, we believe that the idea of Reanalyze (using planning for policy improvement) can be extended to other model-based methods.
>
> > Q3. How to choose the balancing weights $λ_d$ and $λ_{bc}$ in Eq. (7)?
>
> The balancing weights are determined by a hyperparameter search. We find that $λ_d$ = 0.1 and $λ_{bc}$ = 0.01 works quite well across the tasks.

---

### Official Review · Reviewer_bqbS · 2022-07-12

**Rating:** 7
**Confidence:** 3
**Soundness:** 3 good
**Presentation:** 2 fair
**Contribution:** 3 good

**Summary:**

The paper proposes a method to improve sample efficiency of the model-based imitation learning. The method consists in combining three components:
* Efficient-Zero algorithm
* GAIL-style imitation learning approach which will learn a discriminator and provide it to Efficient-Zero algorithm to specify the reward
* BC algorithm to learn a policy which will regularize the acting distribution of Efficient-Zero

The method demonstrates improved data efficiency (in terms of online rollouts) compared to the baseline, and is conceptually simple.

**Questions:**

* Why is it important to use EfficientZero? Can we use any other model-based method?
* Can it be that the main benefit comes from using EfficientZero as the algorithm? What if we use this algorithm in other model-based imitation learning baselines?
* Why did the author choose a specific number of trajectories? How does the method & baselines behave when different number of trajectories are considered?
* Can you provide more intuition / evidence on why each component of the algorithm is important ? (I.e., GAIL, EfficientZero, BC)


**Ethics Review Area:**

["I don’t know"]

**Limitations:**

I think the authors adequately presented the limitations.

**Strengths And Weaknesses:**

Strenghts:
* A method which combines different tricks and achieves a competitive performance in a model-based imitation learning setting
* Conceptual simplicity of the method
* The paper studies an important problem - model-based imitation learning

Weaknesses:
* The paper made multiple somehow arbitrary choices which led to a quite an efficient algorithm, but lacks an intuition on why each of the choices was important. For example, the proposed technique (GAIL & BC component) can be plugged-and-played to any other model-based method as well as non model-based method. Is EfficientZero algorithm really crucial for this technique to work? The paper would benefit from more evidence on why each of the proposed components are important. What would happen if instead of model-based algorithm, we use a model-free algorithm with similar acting distribution (mixture between acting and BC policies), having an additional term to optimize for the GAIL-learned reward function?
* Can it be that baseline methods can be significantly improved if EfficientZero algorithm used in these? Can it be that the main benefit in this work comes from using EfficientZero as the main RL algorithm?
* The authors somewhat arbitrary choose the number of expert trajectories. Why exactly these numbers? How do baseline methods perform when more or less expert demonstrations are available?

---

> ### Author Response · Authors · 2022-08-02
> **Thanks for the review!**
>
> Thanks for the thoughtful review. We address your concerns below. We are happy to discuss any additional questions or concerns.
>
> > Q1. Why is it important to use EfficientZero? Can we use any other model-based method?
>
> Thanks for the question. One important reason we use EfficientZero here is that it comes with a MCTS planning component, which leads to a natural approach to unifying BC and AIL. We notice that MCTS planning has a very unique benefit. It enables our method to use BC actions in a clever way: it will choose BC actions when they are right (leading to long-term distribution matching) and discard them when they are wrong. Model-free methods do not have such capability, since they cannot compute the long-term effects of BC actions via extensive search. Our experiments (Line 301-310) also show that when we reduce the MCTS search, the performance will degrade significantly. Moreover, the other model-based methods like Dreamer also do not have such a benefit. Take dreamer as an example, although it also builds a model of the environment, it still uses model-free methods to learn the policy function with the model. Thus, other model-based approach like Dreamer cannot combine the BC as we did. In conclusion, what makes EfficientZero (MCTS planning) unique and important here is that it can evaluate the effect of BC actions, and use them to speed up learning. Such a feature does not present in model-free and the other latest model-based methods.
>
> > Q2. Can it be that the main benefit comes from using EfficientZero as the algorithm? What if we use this algorithm in other model-based imitation learning baselines?
>
> We agree that EfficientZero does offer some benefits: compared with other model-based imitation learning baselines, its extra MCTS planning component can ensure more effective policy and value updates. However, the benefit also comes from BC. The BC component can provide a very good initial solution to EfficientZero to speed up the learning process, and help the policy escape from local minimum in challenging tasks with high dimensional states. Since the other model-based imitation learning baselines are not based on MCTS, we cannot implement EfficientImitate in the other baselines directly.
>
> > Q3. Why did the author choose a specific number of trajectories? How does the method & baselines behave when different number of trajectories are considered?
>
> Thanks for pointing out this. We have just carried out experiments with fewer trajectories to increase the difficulty. We reduce the number of expert trajectories from 5 to 2 in the state-based experiments, and from 20 to 10 in the image-based experiments. The results are shown as below.
>
> | Method | Cheetah (State) | Walker (State) | Cheetah (Image) | Walker (Image) |
> |  ----  | ----  | ----  | ----  | ----  |
> | BC | 0.50 | 0.12 | 0.33 | 0.11 |
> | DAC | 0.18 ± 0.02 | 0.22 ± 0.03 | 0.04 ± 0.01 | 0.10 ± 0.02 |
> | SQIL | 0.04 ± 0.01 | 0.10 ± 0.03 | 0.05 ± 0.00 | 0.27 ± 0.04 |
> | ValueDICE | 0.43 ± 0.05 | 0.50 ± 0.06 | 0.04 ± 0.01 | 0.06 ± 0.01 |
> | VMAIL | N/A | N/A | 0.12 ± 0.03 | 0.22 ± 0.06 |
> | Ours | **0.94 ± 0.03** | **0.97 ± 0.02** | **0.91 ± 0.02** | **0.93 ± 0.03** |
>
> The performance of our method only degrades a little bit and approximately remains the same level. It still outperforms the baselines by a large margin.
>
> > Q4. Can you provide more intuition / evidence on why each component of the algorithm is important ? (i.e., GAIL, EfficientZero, BC)
>
> Importance of EfficientZero: It provides an MCTS-based RL framework, over which we can unify the two classes of imitation learning algorithms, BC and AIL, with planning.
>
> Importance of BC: It improves the sample efficiency, and helps the policy to escape from the local minimum by giving good candidate actions when we use MCTS to solve the imitation problem.
>
> Importance of GAIL: It defines the goal of MCTS planning, which is to ensure long-term distribution matching. We do agree that GAIL can be replaced with its improved versions (Line 179-183). One reason we use GAIL here is that it is the vanilla AIL algorithm. Using the improved AIL/IRL algorithms here might further improve the performance and this is left for future work.

---

> > ### Comment · Reviewer_bqbS · 2022-08-09
> > **Thank you for the response**
> >
> > Thank you for the response.
> >
> > Your explanations make sense. I would encourage you to propagate them to the paper in order to help the readers also to build up the intuitions that you have.
> >
> > Thank you for conducting additional experiments.

---

### Meta-Review · Area_Chair_gdhr · 2022-08-23

**Recommendation:** Accept
**Confidence:** Certain

**Metareview:**

This paper introduces a simple approach that improves the sample efficiency of model-based RL for continuous control tasks.  The proposed approach, EfficientImitate builds on EfficientZero and uses a hybrid BC-AIL training scheme.  The contribution is relatively simple and is shown through satisfactory experiments to give a substantial sample efficiency boost.  The paper is clear and appropriately contextualizes its contribution.  All reviewers found the paper to be clear, novel, technically sound, and empirically well validated.  The results show that the innovations constitute a meaningful contribution to a fairly general problem class.

In initial reviews, two of the three reviewers mentioned that the method is not shown on discrete-action problems.  Personally, I wouldn't have seen this to be a major concern, since continuous control problems are a large problem class.  However, the authors replied with a comment indicating that for lunar lander (a discrete action atari environment), the method works.  It is unclear if this addition was cherry-picked among discrete environments and it is also unclear to me if the authors will add this to the paper.  Nevertheless, as noted, I don't find this to be a major gap in the paper as it currently stands.

Given the sufficiently positive reviews, level of review agreement, and my own reading, I endorse this paper for acceptance.

**Award:**

No

---

### Decision · Program_Chairs · 2022-09-14

Accept